# Effectiveness of the Strengthening Families Programme in the UK at preventing substance misuse in 10–14 year-olds: a pragmatic randomised controlled trial

Jeremy Segrott [1,2] David Gillespie [1] Mandy Lau [1] Jo Holliday [3]
Simon Murphy [2] David Foxcroft [4] Kerenza Hood [1]
Jonathan Scourfield [5] Ceri Phillips [6] Zoe Roberts,[7] Heather Rothwell,[2]
Claire Hurlow,[8] Laurence Moore [9]

For numbered affiliations see end of article.

**Correspondence to**
Dr Jeremy Segrott;
SegrottJ@cardiff.ac.uk

## ABSTRACT

**Objectives** The Strengthening Families Programme 10–14 (SFP10-14) is a USA-developed universal group-based intervention aiming to prevent substance misuse by strengthening protective factors within the family. This study evaluated a proportionate universal implementation of the adapted UK version (SFP10-14UK) which brought together families identified as likely/not likely to experience/present challenges within a group setting.

**Design** Pragmatic cluster-randomised controlled effectiveness trial, with families as the unit of randomisation and embedded process and economic evaluations.

**Setting** The study took place in seven counties of Wales, UK.

**Participants** 715 families (919 parents/carers, 931 young people) were randomised.

**Interventions** Families randomised to the intervention arm received the SFP10-14 comprising seven weekly sessions. Families in intervention and control arms received existing services as normal.

**Outcome measures** Primary outcomes were the number of occasions young people reported drinking alcohol in the last 30 days; and drunkenness during the same period, dichotomised as 'never' and '1–2 times or more'. Secondary outcomes examined alcohol/tobacco/substance behaviours including: cannabis use; weekly smoking (validated by salivary cotinine measures); age of alcohol initiation; frequency of drinking >5 drinks in a row; frequency of different types of alcoholic drinks; alcohol-related problems. Retention: primary analysis included 746 young people (80.1%) (alcohol consumption) and 732 young people (78.6%) (drunkenness).

**Results** There was no evidence of statistically significant between-group differences 2 years after randomisation for primary outcomes (young people's alcohol consumption in the last 30 days adjusted OR=1.11, 95% CI 0.72 to 1.71, p=0.646; drunkenness in the last 30 days adjusted OR=1.46, 95% CI 0.83 to 2.55, p=0.185). There were no statistically significant between-group differences for other substance use outcomes, or those relating to well-being/stress, and emotional/behavioural problems.

**Conclusions** Previous evidence of effectiveness was not replicated. Findings highlight the importance of evaluating interventions when they are adapted for new settings.

**Trial registration number** ISRCTN63550893.Cite Now

## Strengths and limitations of this study

► Both trial arms had good participant retention rates at 2-year follow-up.
► Recruitment, intervention delivery systems and implementation replicated how the intervention would be delivered outside a trial setting, thus maximising applicability of findings.
► There were good levels of agreement between self-reported smoking behaviour and cotinine levels in collected saliva samples.
► Most outcomes, including the primary outcomes relating to alcohol, could only be measured using self-report.

## INTRODUCTION

Prevention of substance misuse among young people is a key public health priority internationally. Early initiation of alcohol consumption and drug use during adolescence is associated with short-term negative impacts (eg, accidental injury, educational attainment) and poorer health over the life course, including substance use disorders and mental health problems. Family-based risk and protective factors, encompassing parental modelling of alcohol use,[1] substance use rules and monitoring,[2] and parenting styles and family relationships, have been identified as important,[3] and influence a range of outcomes alongside substance use such as

adolescent mental health.[4] This has led to considerable policy interest in developing prevention interventions targeting key parenting/family variables[5] and a need to develop the international evidence base.[6]

The Strengthening Families Programme 10–14 (SFP10-14) is a USA-developed universal group-based intervention that aims to prevent substance misuse in young people by strengthening protective factors within the family.[7]

Three main trials of SFP10-14 in the USA using a universal delivery model have been conducted—each led by the team responsible for intervention development at Iowa State University (ISU). These trials have demonstrated the programme's effectiveness in delaying the onset of alcohol use, reducing uptake of smoking, the incidence of harder drug use (methamphetamine)[8 9] (6.5 years past baseline) and substance use at 10-year follow-up.[10] However, findings from the trials conducted in the USA have been criticised in relation to data analysis and reporting practices, sample 'selection bias' and low rates of intervention attendance.[11 12] For example, Gorman's methodological critique of the ISU's Capable Families and Youth evaluation argues that:

> … there were no statistically significant results reported using 2-tailed tests for individual point-in-time measures of alcohol, cigarettes and marijuana use, or for the drunkenness measure, in the CaFaY evaluation. The five effects reported as statistically significant were based on 1-tailed tests of significance and one of these was for a measure unique to that publication and two were based on a highly irregular practice of using posttest data as the baseline.[12] (p 35)

A critique by the same author of the third ISU-led trial, PROSPER (PROmoting School-community-university Partnerships to Enhance Resilience), notes that only 19% of young people allocated to the intervention attended at least one SFP10-14 session. Another critique of this study is that it fails to provide adequate details concerning rates of refusal by participants to provide data at 7.5 years of follow-up, and uncertainty regarding how many of those in the intervention arm who did complete questionnaires had attended any SFP10-14 session.[11]

The current study evaluates the version of the intervention that was culturally adapted for the UK (SFP10-14UK). It seeks to address some of the key limitations of the ISU-led trials. Steps to achieve this include prespecification of outcomes and analysis plans, prospective publication of a trial protocol and comprehensive reporting of data on recruitment, retention and intervention attendance rates.

Universal interventions (provided without recourse to criteria relating to risk/need) may inadvertently perpetuate inequalities because of their sometimes low reach among those most at need.[13] Targeted interventions may also face difficulties in identifying and recruiting such groups, create barriers to engagement through stigmatisation of potential participants[14] and generate iatrogenic effects through reinforcement and normalisation of harmful norms.[15 16] Targeted interventions, which bring together families with high levels of need, may also present problems in relation to feasibility of delivering activities with fidelity.[17] A distinctive aspect of this trial is that the intervention was implemented using a proportionate universal (PU) approach—combining universal provision but with its intensity/scale varied according to need.[18] This aimed to address some of the limitations of both universal and targeted approaches described above and operationalise delivery of a universal intervention within an existing UK system mainly geared towards targeted interventions. Driven by a desire to address health and social inequalities, a PU approach provides services universally but 'at a scale and intensity of action […] proportionate to the level of disadvantage', thus aiming to improve engagement with, and effectiveness among, groups with higher needs[18] (p 16). Universal provision is designed to achieve sustainability, secure political and financial support and avoid stigmatisation.[19] Variation in the 'scale and intensity' of interventions is applied to recruitment (greater effort/resources may be needed to engage certain social groups) and during programme implementation to meet differing needs, aiding participant retention. Thus, within a PU approach, targeting may comprise aspects of both *selectivism* (identifying whether certain groups should be targeted and how) and *particularism*—how interventions can be tailored to meet differing needs.[20]

The PU approach adopted had two key components. First, at recruitment stage the intervention was provided on a universal basis but with greater efforts and resources dedicated to recruiting those families with higher level support needs. This recruitment strategy aimed to reach families with higher level needs while avoiding creating barriers to engagement, and the limitations of a purely targeted approach in identifying such families.

Second, groups of families brought together for the intervention were composed of approximately 30% who were likely to experience or present challenges within a group setting—for example, young people not attending or excluded from school/class/year group or working with behavioural support teams; autistic spectrum condition; aggression/anger; low literacy skills; learning difficulties; mental health issues; English/Welsh not being a family's first language; and substance misuse (parent or young person), and 70% without such challenges. In line with a PU approach, this group composition was designed to ensure that there were sufficient numbers of staff to provide additional support to families who might experience or present challenges within a group setting. It was also designed to enable delivery of activities as intended within a highly manualised and structured intervention, and to promote prosocial group dynamics, with the goal of facilitating behaviour change processes hypothesised to generate the intervention's intended impacts.

## Aim

The trial's primary objective was to assess the effectiveness of SFP10-14UK in preventing alcohol misuse in young people. Its secondary objectives were to assess the programme's impact on drug misuse, smoking behaviour, alcohol initiation and drink-related problems and school performance, among young people. Tertiary objectives were to measure the extent to which SFP10-14UK had effects on young people's mental health and well-being, and protective factors for alcohol and tobacco use/misuse (eg, family functioning, parenting and young people's peer pressure resistance skills). The trial encompassed a cost-consequences analysis to assess whether the intervention was cost-effective. It assessed if there were important variations in delivery and receipt across trial sites, and identified key programme theory, content and processes.

## METHODS

### Study design

The trial comprised a pragmatic cluster-randomised controlled effectiveness trial with families as the unit of randomisation, and all individuals within a family recruited into the study allocated to the same arm. There were embedded process and economic evaluations. The study took place in seven counties of Wales, UK. Full details of the design, methods and outcomes have been published in the study protocol.[21] Programme delivery systems, including staffing arrangements and participant eligibility, were designed to mirror future 'real world' implementation outside of a research context to maximise generalisability of findings. The researchers did not impose any additional standardisation of intervention delivery. Local authorities and third sector organisations with responsibility for supporting families or substance misuse prevention led multiple agency partnerships that delivered the programme in community settings. The trial was prospectively registered with the ISRCTN registry on 3 December 2009, and prior to any participants being enrolled.

### Intervention

Drawing on theoretical models relating to biopsychosocial vulnerability, youth resiliency and family process, SFP10-14 addresses risk and protective factors within the family environment known to shape later substance use behaviours.[7] It comprises seven weekly sessions each lasting 2 hours. In the first hour, parallel groups of young people and parents/carers from up to 12 families develop their understanding and skills, led by trained facilitators. Each session is guided by a facilitator manual which details activities and how they are to be delivered. In the second hour, parents/carers and young people come together in family units to practise key skills. The programme has been culturally adapted for the UK (SFP10-14UK). Adaptations comprised changes to language and terms to fit with a UK context, and production of revised videos which used 'real life' situations filmed in home settings instead

of studio-based narration. Core content and functions of the intervention were retained and not subjected to significant change.[22] A logic model for the culturally adapted intervention has been developed by the authors.[23]

### Participants

Each programme was open to families with young people aged 10–14 from a large geographical area (eg, town or rural district). Self-referrals to the Programme came forward in response to awareness raising in community and educational settings. This included distribution of fliers to parents across relevant school year groups, attendance at community events, promotion via social media and placing of leaflets in local venues such as leisure centres. Other referrals came from agencies such as education, health and social services which identified families that may benefit from participating in the Programme. Potential referrers and families were informed that the Programme was being run as part of a trial and were provided with information about the trial. When a family was referred or applied to the SFP10-14UK they were visited by a member of the programme delivery team who undertook a needs and eligibility assessment, which included exploration of any difficulties or support needs which participants might experience within a group setting. For example, this encompassed exploring the support needs of those with low literacy levels (as much of the programme involved text-based materials) or addressing how conflict within families might manifest when they came to work together on activities during sessions. Based on the information contained in the family referral/application form and the needs and eligibility assessment, the programme delivery coordinator determined if eligible families were likely to experience or present challenges within a group setting. If families were deemed eligible to attend the programme they were asked if they were willing for a member of the research team to have access to their referral notes and to visit them. Where the family agreed, a research fieldworker visited them to obtain informed consent to participate in the trial and collect baseline data using a computer-assisted personal interview.

Any family deemed eligible to attend SFP10-14UK was included in the research trial, subject to their giving consent. Trial inclusion and exclusion criteria are listed in table 1. Informed consent was obtained from all participants. For a family to be included in the research trial, at least one parent/carer needed to consent to participate, and parent/carer consent for the inclusion of at least one young person in the trial was required. Young people also gave consent for their participation in the trial.

### Randomisation and masking procedures

Remote computerised randomisation was conducted by the research fieldworker at the end of the baseline interview once consent and baseline data had been obtained. Families were randomised within strata defined by the seven areas in which the study was conducted and

**Table 1** Trial inclusion and exclusion criteria

| Inclusion criteria | Exclusion criteria |
|---|---|
| Families in which at least one parent/carer and one young person are willing to attend the programme together. | Families in which either a parent or young person does not want to attend the programme. |
| Families with the ability to speak English (help can be provided for parents or young people with low literacy levels). Some programmes may also be delivered through the medium of Welsh if there is sufficient demand. | Families in which there are either parents or young people who cannot speak English (or Welsh, where appropriate). |
| Families where a programme is being offered at a location to which it is practicable for a family to travel (as determined by the programme coordinator) within the next 3 months. | Families where there is no programme being offered at a location to which it is practicable for a family to travel (as determined by the programme coordinator) within the next 3 months. In such a case, the family would not be excluded. They will be placed on a waiting list for the programme and will be contacted when a programme is available. They will then be recruited into the trial. |
| Families with a young person aged 10–14. | – |
| – | Families who do not live together, for example, the child/children are in care. |
| – | Families with very high needs or challenges (such as serious substance misuse problems, family breakdown or crisis). |

using minimisation with a random element set at 80%. The minimisation algorithm aimed to achieve balance between randomised groups on family categorisation (family without challenge/family with challenges in a group setting), the average age of the young people within a family who were recruited into the trial (<12/12+) and the number of young people within a family who were recruited into the trial (1/>1). Families randomised to the intervention arm received SFP10-14UK in addition to usually provided services. Those randomised to the control continued to receive usually provided services only. These services included local authority parenting and family services, school-based support and other providers' services such as substance misuse prevention teams, child and adolescent mental health services (CAMHS) and programmes operated by charities. There was some variation in the provision and organisation of services across the trial areas. Data on families' health, social care, education and criminal justice service utilisation were collected from parents/carers at 9, 15 and 24-month follow-up points. Due to the nature of the intervention, it was not possible for participants or those delivering the intervention to be blinded to families' group assignment. Assessment of trial outcomes was done blind to group assignment.

## Outcomes

The trial had two primary outcomes: the number of occasions that young people reported having drunk alcohol in the last 30 days; and drunkenness during the last 30 days, dichotomised as 'never' and '1–2 times or more'.

Secondary outcomes were: reported use of cannabis (ever vs never); weekly smoking (yes vs no, validated by salivary cotinine measures); age of alcohol use initiation; frequency of drinking more than five drinks in a row in the last 30 days; frequency of different types of alcoholic drinks; alcohol-related problems; and General Certificate of Secondary Education (GCSE) performance at age 15/16 (number of GCSEs passed and grades achieved, measured as a continuous outcome). All primary and secondary outcomes were collected from children at 2-year follow-up, with the exception of GCSE results, which it was proposed to collect via the Secure Anonymised Information Linkage Databank,[24] once all participants had completed GCSEs/left compulsory education.

The trial's tertiary outcomes were: parenting (General Child Management Scale child report); family functioning (Family Relationship Index); children's well-being and stress (Strengths and Difficulties Questionnaire (SDQ) scores); children's health status; Short Form Health Survey (SF-36); parents/carers' health status (General Health Questionnaire and EuroQol-5 Dimension (EQ-5D)); costs and an assessment of relative cost-effectiveness, derived from a cost-consequences analysis; children's smoking status (ie, whether they have ever smoked/smoked now); young people's self-efficacy; age of first cigarette; and age of first drug use. Online supplemental tables S1 and S2 provide a description of the trial's primary, secondary and tertiary outcomes, and the specific measures used at main follow-up at 24 months.

Selection of outcomes and length of follow-up were driven by the intervention's hypothesised mechanisms of action and impacts, including long-term substance use prevention, and short/medium-term mediators, particularly family relationships and parenting.

Procedures were put in place to assess and respond to reports of adverse events. Serious adverse events (SAEs) were reviewed by a senior member of the trial team within 24 hours to assess the nature of the SAE, its seriousness,

causality and expectedness. Where an SAE was both related (resulting from administration of any of the research procedures) and unexpected, the trial manager notified the research ethics committee (REC) within 15 days of receiving notification of the SAE.

## Cost-consequences analysis

The costs of the programme were identified, measured and valued in monetary terms and combined with changes in resource utilisation of services as a result of programme participation. Distinctions were made between the costs incurred in each programme area to assess variation and potential for efficiency gains. The costs of delivering the programme were derived from conversations with relevant colleagues and comprised staff time (plus training costs), venue and equipment costs, provision of support facilities, promotional materials and other resources involved in family recruitment, together with costs incurred by participants. The cost per participant and cost per family were computed for the study and for each of the areas of implementation. The extent to which participation resulted in changes in utilisation of services was assessed by undertaking a series of participant interviews. Changes to resource utilisation were translated into monetary terms using appropriate published unit cost data and merged with the costs of programme delivery to generate the net cost of the programme, net cost per family, net cost per adult and net cost per young person. The net cost of delivering the programme was used alongside differences in primary, secondary and tertiary outcomes to generate a set of ratios of relative cost-effectiveness within the study period. Sensitivity analyses were undertaken to assess the extent to which changes in parameter estimates affected the baseline findings.

## Statistical analysis

We aimed to randomise 756 families in total (378 per arm). Our target sample size provides 80% power with a two-sided alpha at 0.025 (halved to account for two primary outcomes) to detect either a 12 percentage point difference in young people reporting having drunk alcohol at the 24-month follow-up point (assuming a control group prevalence of 48%)[25] or a 10 percentage point difference in young people reporting having been drunk at the 24-month follow-up point (assuming a control group prevalence of 22%).[25] The sample size was inflated based on the average family including 1.25 young people in the trial, an intracluster correlation coefficient (ICC) of 0.2 and 25% loss to follow-up.

Descriptive statistics were calculated using means (SD), medians (IQR) and percentages as appropriate.

The primary analyses were conducted on a modified intention-to-treat population (participants who provided outcome data analysed in the arm to which they were randomised) and involved fitting two-level logistic regression models to whether or not young people reported drinking alcohol/being drunk in the 30 days prior to their 24-month interview, with responses from young people nested within families. The models controlled for the variables balanced on at randomisation (other than number of young people, as this was balanced on to ensure an equal number of young people per group, rather than scientific reasons), the study area and the baseline-reported alcohol use/drunkenness (depending on the outcome). To test the strength of the findings of the primary analyses, the models were reanalysed controlling for gender, time from randomisation to 24-month follow-up and without controlling for any covariates (preplanned analyses). The primary outcomes were also analysed as ordinal variables (ie, the number of occasions in the last 30 days that young people reported drinking alcohol/being drunk). The three categories used in the analysis were 'no occasions', '1–2 occasions' and '3 or more occasions'. Two-level ordinal regression models were fitted to both outcomes, with and without adjusting for variables balanced on at randomisation and baseline alcohol use/drunkenness.

In order to assess the impact that intervention non-receipt had on the findings of the primary analyses, double logistic structural mean models using generalised method of moments were fitted to each primary outcome.[26] A binary definition of intervention receipt was agreed by the study team a priori which involved participants attending at least five sessions, without missing more than one session in a row. The quantitative measure of intervention receipt was also used (ie, the number of weeks attended).

Multiple imputation was used in a sensitivity analysis to assess the impact of missing primary outcome data on the conclusions drawn. Additional sensitivity analyses were also performed using more extreme assumptions about missing primary outcome data. For further details, see the online supplemental material.

Secondary outcomes were analysed by fitting two-level logistic regression models (proportion of young people at 24 months classing themselves as weekly smokers, reporting having ever used cannabis, having used cannabis in the previous 12 months, having used cannabis in the previous 30 days and having experienced at least one of the listed alcohol-related problems in the previous 12 months) and two-level Cox proportional hazards models (time to initiation of alcohol, smoking and drugs—in those participants who had not initiated at baseline). Covariate adjustment for secondary analyses mirrored that used for the primary analyses.

Potential modifiers of the intervention with respect to both primary outcomes were investigated by fitting interaction terms to trial arm and preplanned subgroups (age, gender and smoking behaviour of the young person, drinking behaviour of the parents/carers, socioeconomic status assessed using the Family Affluence Scale (low, medium or high) and highest occupational status of parents/carers (categories based on the National Statistics Socio-economic classification (NS-SEC) self-coded method), family categorisation (family likely to experience challenges in a group setting/family without

challenges in a group setting), young person-reported SDQ, domains of the Family Relationship Index and averaged young person and parent/carer scores of General Child Management Scale at baseline).

As it was one of the distinguishing features of this trial, further exploratory post hoc subgroup analyses were conducted to investigate potential differential intervention effects by family categorisation for the smoking, cannabis use and SDQ outcomes, the latter being included because of its focus on mental health, conduct problems and peer influences.

All analyses were conducted using STATA V.13.0.

### Role of funder
The study funder was not involved in the design of the study, or collection, analysis, or interpretation of the data. The corresponding author had access to all data in the study and had final responsibility for the decision to submit for publication.

### Patient and public involvement
Members of the public were involved in refining and piloting of outcome measures and study procedures. This included a mix of families who had already received SFP10-14 (and who were able to comment on the perceived relevance of proposed outcomes) and those without any prior knowledge of the intervention (and were therefore able to provide input on how trial participants with no previous knowledge might perceive study materials). Piloting comprised asking participants to complete baseline interviews with researchers, and then provide feedback on clarity of written information, time taken to complete the baseline interview and the acceptability and appropriateness of outcome measures. This information was then used to refine the final selection of outcome measures and format of information sheets, etc. Input was also sought from a young people's research advisory group - Advice Leading to Public Health Advancement (ALPHA), led by a public involvement officer embedded within the study team. The public involvement officer established stakeholder groups—formed of trial participants, to advise the study team at key points on recruitment and retention strategies, maintaining contact with the trial cohort and plans for disseminating trial findings.

### RESULTS
Participant recruitment took place between 8 February 2010 and 18 June 2012. Seven hundred and fifteen families were recruited into the trial. This comprised 918 parents/carers and 931 young people. The number of participants completing questionnaires at each assessment point is included in the trial profile (figure 1). Primary outcome data were collected from young people. The primary analyses included 746 young people for the alcohol use outcome at 2 years after randomisation (80.1%) and 732 for the drunkenness outcome at 2 years after randomisation (78.6%). Completion rates were similar across trial

arms. Analysis of levels of service utilisation data at 9, 15 and 24-month follow-up did not identify any discernible differences between the intervention and control arms.

### Baseline characteristics
Table 2 illustrates the characteristics of parents/carers and young people at baseline in control and intervention groups. Randomised groups were broadly similar at baseline, with small differences noted for young person gender and parents/carers' reported cigarette use.

### Primary analysis
Table 3 shows the results of alcohol use and drunkenness for young people in the previous 30 days prior to their 24-month follow-up interview . Young people allocated to the SFP group had higher alcohol use and drunkenness prevalence than those allocated to the control group (26.4% vs 24.6% for alcohol use, 10.2% vs 8.3% for drunkenness). However, results from the multilevel models indicated insufficient evidence to reject the null hypothesis of no difference between groups (alcohol use adjusted OR (AOR)=1.11, 95% CI 0.72 to 1.71, p=0.646; drunkenness AOR=1.46, 95% CI 0.83 to 2.55, p=0.185). Conclusions were unaffected by preplanned secondary analyses of the primary outcomes (see online supplemental tables S3–S5).

### Subgroup analysis for primary outcomes
Online supplemental tables S6 and S7 provide the results of the planned subgroup analysis for alcohol use and drunkenness in the previous 30 days. There was no evidence of statistically significant differential intervention effects for any of the subgroups for either primary outcome.

### Other substance use secondary outcomes
There was no evidence of statistically significant differences between groups for the other substance use outcomes (table 4). Rates of agreement between self-reported weekly smoking behaviour and saliva cotinine are shown in online supplemental table S8.

### GCSE performance at age 15/16
Based on the analysis of the primary and secondary self-reported outcomes at 2-year follow-up, it was determined that collection of the GCSE data was not justified.

### Post hoc subgroup analysis
Table 5 provides the results of unplanned subgroup analysis aiming to explore differential intervention effects on the secondary substance use outcomes and SDQ score by whether or not the family was likely to experience/present challenges in a group setting. The motivation for this exploratory work stemmed from the subgroup findings for the two primary outcomes, which demonstrated small (not statistically significant) differential effects favouring families experiencing/presenting challenges within a group setting. The attrition rate for the 24-month data was similar between families with or without challenges

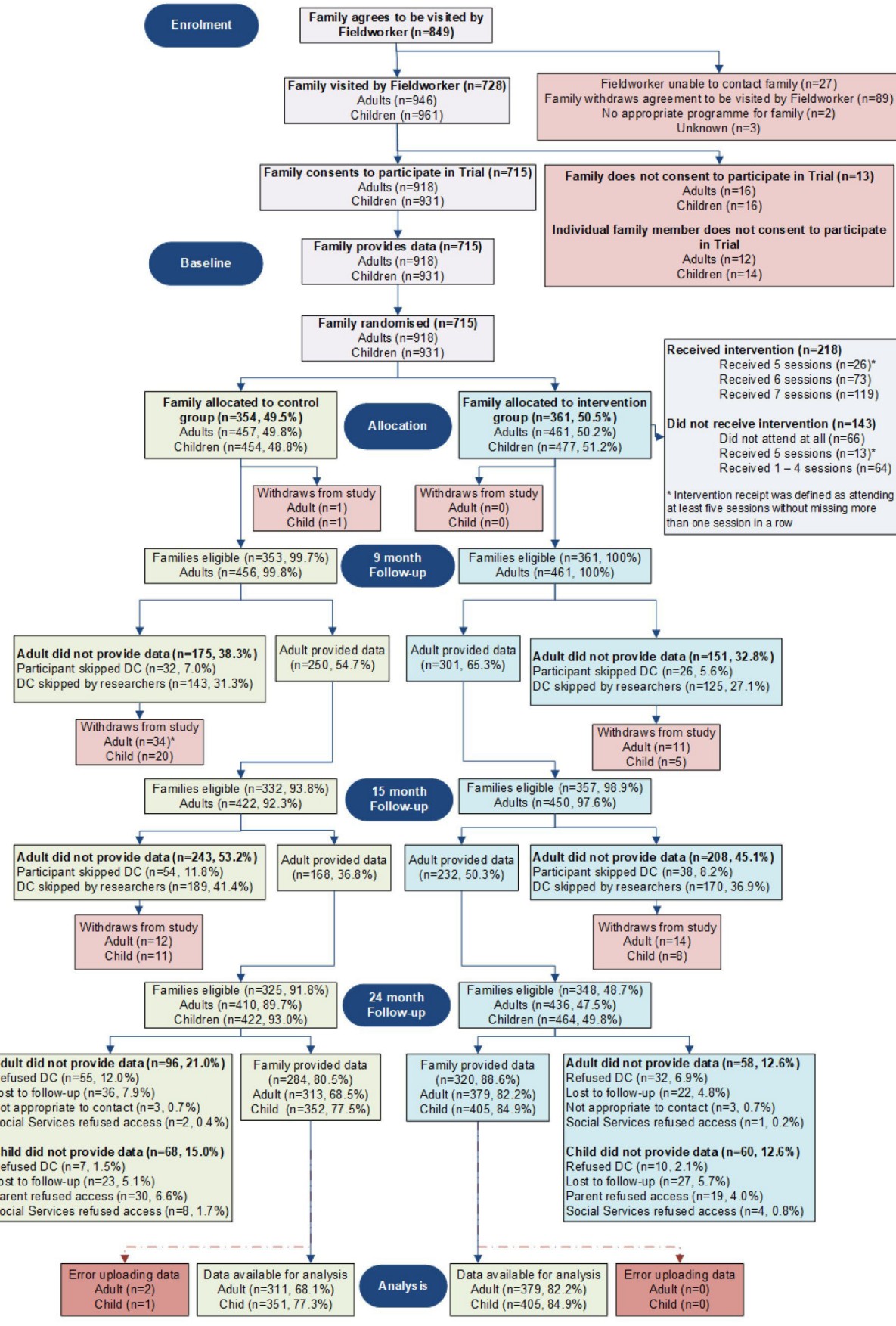

**Figure 1**  Trial consort participant flow chart.

*Thirty-two adults in the control group withdrew at the 9-month data sweep. A further two adults withdrew prior to being scheduled for their 15-month data sweep.

**Table 2** Baseline characteristics for parents/carers and young people according to experimental group

| Parent/carer baseline characteristics | | Control (n=457) | SFP (intervention) (n=461) |
|---|---|---|---|
| Age (median, IQR) | | 37.0 (32.0–43.0) | 37.0 (33.0–42.5) |
| Gender | Female | 105 (23.0%) | 103 (22.3%) |
| | Male | 352 (77.0%) | 358 (77.7%) |
| Ethnic group | White | 451 (98.7%) | 458 (99.6%) |
| | Non-white | 6 (1.3%) | 2 (0.4%) |
| Relationship status | Single, not in a relationship | 54 (12.0%) | 68 (14.8%) |
| | Single, in a relationship | 97 (21.5%) | 91 (19.8%) |
| | Married/civil partnership | 200 (44.3%) | 183 (39.9%) |
| | Separated/divorced | 95 (21.1%) | 113 (24.6%) |
| | Widowed | 4 (0.9%) | 3 (0.7%) |
| | Other | 1 (0.2%) | 1 (0.2%) |
| Ever smoked a cigarette | Never | 231 (50.5%) | 207 (44.9%) |
| | Yes, at least once | 226 (49.5%) | 254 (55.1%) |
| Ever drunk alcohol | Never | 85 (18.6%) | 90 (19.5%) |
| | Yes, at least once | 372 (81.4%) | 371 (80.5%) |
| AUDIT-C (Alcohol Use Disorder Identification Test for Consumption) | Low risk | 304 (66.5%) | 294 (64.6%) |
| | High risk | 153 (33.5%) | 161 (35.4%) |
| Ever taken drug | Never | 300 (65.6%) | 307 (66.6%) |
| | Yes, at least once | 157 (34.4%) | 154 (33.4%) |
| Deprivation overall score (median, IQR) | | 27.5 (18.5–41.2) | 29.4 (18.0–44.5) |

| Young people baseline characteristics | | Control (n=454) | SFP (n=477) |
|---|---|---|---|
| Age (median, IQR) | | 11.5 (10.0–13.0) | 12.0 (11.0–13.0) |
| Gender | Female | 231 (50.9%) | 269 (56.4%) |
| | Male | 223 (49.1%) | 208 (43.6%) |
| Ethnic group | White | 446 (98.5%) | 474 (99.6%) |
| | Non-white | 7 (1.5%) | 2 (0.4%) |
| Family affluence scale | Low (0–2) | 83 (18.4%) | 99 (20.9%) |
| | Medium (3–5) | 234 (52.0%) | 231 (48.8%) |
| | High (6–9) | 133 (29.6%) | 143 (30.2%) |
| Ever smoked a cigarette | Never | 330 (72.7%) | 348 (73.0%) |
| | Yes, at least once | 124 (27.3%) | 129 (27.0%) |
| Ever drunk alcohol | Never | 314 (69.3%) | 326 (68.5%) |
| | Yes, at least once | 139 (30.7%) | 150 (31.5%) |
| Ever taken drugs | Never | 433 (95.4%) | 450 (94.3%) |
| | Yes, at least once | 21 (4.6%) | 27 (5.7%) |
| Child deprivation overall score (median, IQR) | | 28.1 (19.7–39.2) | 28.2 (18.3–39.6) |

SFP, Strengthening Families Programme.

in a group setting (17.1% for families with challenges vs 12.2% without). The pattern found for the primary outcomes (ie, differential outcomes favouring families with challenges) was reflected across the additional outcomes investigated. Taking the 'cannabis' outcome as one example, we see that the odds of reporting cannabis use in the last 12 months are higher for the main effect (ie, higher for those in a family with challenges and allocated to the control arm, compared with families without challenges in a group setting allocated to the control arm), and the interaction term indicates a reversal of this effect for those in the SFP arm (ie, cannabis use is higher

**Table 3** Primary alcohol use and drunkenness outcomes (24-month follow-up)

| Outcome | ICC* | Proportion of events in control arm (%) | Proportion of events in SFP arm (%) | 95% CI | | | |
| --- | --- | --- | --- | --- | --- | --- | --- |
| | | | | Adjusted OR† | Lower limit | Upper limit | P value |
| Alcohol use in previous 30 days‡ | 0.25 | 85/345 (24.6) | 106/401 (26.4) | 1.11 | 0.72 | 1.71 | 0.646 |
| Drunkenness in previous 30 days§ | 0.18 | 28/338 (8.3) | 40/394 (10.2) | 1.46 | 0.83 | 2.55 | 0.185 |

*Intracluster correlation coefficient: proportion of the total variance (in whether or not a young person reports having used alcohol/been drunk in the 30 days prior to their 24-month follow-up interview) attributable to differences between families. Calculated using the standard $\pi^2/3$ estimator for binary outcomes.
†Analyses adjusted for variables that were balanced on at randomisation (study area, family categorisation and average age of young people within the family) and baseline alcohol use/drunkenness (corresponding with the outcome).
‡Number of occasions in the last 30 days that participant has drunk alcohol (asked at 24-month follow-up)—dichotomised to 0/1 or more occasions.
§Number of occasions in the last 30 days that participant has been drunk (asked at 24-month follow-up)—dichotomised to 0/1 or more occasions.
SFP, Strengthening Families Programme.

for families without challenges allocated to the SFP arm than families with challenges allocated to the SFP arm).

### Cost-consequences analysis results

The costs associated with the implementation of the intervention were £1 163 278, which equated to £20 773 per programme and £1240 per participant, with wide variation in costs across programme areas. There was no difference in costs of healthcare utilisation between the groups at 9, 15 and 24-month follow-up. At 9 and 15-month follow-up, the main cost driver was that of primary care usage, while at 24-month follow-up the main driver was inpatient episodes. There was no difference between the groups for adult EQ-5D scores at baseline or 24-month follow-up.

EQ-5D scores for adults at both baseline and 24-month follow-up were very similar between the intervention and control arms. Conversely, the EQ-5D scores for the children in the intervention arm saw a difference in the scores between baseline and 24-month follow-up (a mean increase of 0.043 per participant in the intervention arm, compared with 0.030 in the control arm).

A complete case analysis was conducted looking at the number of parents who answered the 9, 15 and 24-month questionnaires and who also answered both the baseline and 24-month EQ-5D questionnaire (intervention n=197; control n=129). The mean EQ-5D difference in the control group was 0.053 at a cost difference of £6271, compared with 0.007 and £7175 in the intervention group. This resulted in a difference of −0.046 at a cost of £904 with an incremental cost-effectiveness ratio of −£19 652. Therefore, the intervention arm was dominated.

As with the vast majority of outcomes, there was no difference in resource utilisation or associated costs between the trial arms at 9, 15 and 24-month follow-up. This was the case for resource utilisation in general and specifically to family/parenting-based support services.

### Process evaluation

Detailed findings from the process evaluation have been reported.[23] Of the 56 programmes run, 47 (84%) achieved the target group size (5–12 families) at enrolment. Facilitators rated participant engagement high in 94% of activities. There was no evidence that engagement differed by group size. Twenty-two (39%) of the 56 programmes achieved the target composition of families with challenges (30%) and those without challenges in a group setting (70%). However, most of the remaining programmes (39/56) achieved a mix of families which was in line with the intended group composition model (ie, the majority of families were those without challenges in a group setting).

Overall implementation fidelity was good. Facilitators were asked to rate the fidelity of the sessions they delivered using a checklist designed by the SFP10-14 developers. Data were available for 50 of 56 programmes, across which facilitators rated 96% of individual activities as mostly/fully covered—this varied across trial sites from 90% to 99%. To assess the reliability of facilitator reports, a sample of 47 sessions from the 50 programmes for which data were available were observed and scored by researcher observers using the same fidelity assessment sheet. Observers rated 77% of observed activities as mostly or fully covered (range 63%–88%): ICC scores from two observers=0.73 (95% CI 0.65 to 0.79). Facilitators' and observers' scores agreed 83% of the time (area range 73%–93%).

Recruitment of families was a key implementation challenge. Practitioners reported that the PU approach heightened such challenges as the universal recruitment diverged from the focus of existing systems and referring agencies on families with higher level needs. However, facilitators identified the PU approach to group composition (with a mix of families with and without challenges within a group setting) as helpful in enabling activities to

**Table 4** Other substance use secondary outcomes (24-month follow-up)

| Outcome | | ICC* | Proportion of events in control arm (%) | Proportion of events in SFP arm (%) | Adjusted OR | 95% CI Lower limit | 95% CI Upper limit | P value |
|---|---|---|---|---|---|---|---|---|
| Alcohol-related problems in the last 12 months | | 0.18 | 51/326 (15.6) | 60/384 (15.6) | 0.95 | 0.59 | 1.54 | 0.849 |
| Currently smoke at least one cigarette a week | | 0.41 | 52/347 (15.0) | 59/402 (14.7) | 0.94 | 0.53 | 1.66 | 0.828 |
| Ever taken cannabis | | | 46/338 (13.6) | 58/393 (14.8) | 1.21 | 0.57 | 2.55 | 0.617 |
| Taken cannabis at least once in the last 12 months | | 0.43 | 41/337 (12.2) | 49/393 (12.5) | 1.01 | 0.51 | 1.97 | 0.985 |
| Taken cannabis at least once in the last 30 days | | | 20/338 (5.9) | 21/393 (5.3) | 0.77 | 0.36 | 1.68 | 0.518 |
| Time to alcohol initiation† | | | 88/246 (35.8) | 81/273 (29.7) | 0.86 | 0.64 | 1.17 | 0.344 |
| Frequency of consuming 5+ drinks in a row in the last 30 days‡ | 0 | 0.00 | 315/345 (91.3) | 348/401 (86.8) | 1.55 | 0.95 | 2.51 | 0.080 |
| | 1–2 | | 15/345 (4.3) | 36/401 (9.0) | | | | |
| | 3–5 | | 15/345 (4.8) | 17/401 (4.2) | | | | |
| Frequency of drinking beer‡ | Never | 0.00 | 290/346 (83.8) | 331/401 (82.5) | 1.15 | 0.77 | 1.72 | 0.492 |
| | Rarely | | 20/346 (5.8) | 17/401 (4.2) | | | | |
| | Every month | | 26/346 (7.5) | 36/401 (9.0) | | | | |
| | At least once a week | | 10/346 (2.9) | 17/401 (4.2) | | | | |
| Frequency of drinking wine‡ | Never | 0.00 | 323/345 (93.6) | 375/402 (93.3) | 1.06 | 0.59 | 1.92 | 0.837 |
| | Rarely | | 5/345 (1.4) | 7/402 (1.7) | | | | |
| | At least once a month | | 17/345 (4.9) | 20/402 (5.0) | | | | |
| Frequency of drinking spirits‡ | Never | 0.71 | 298/346 (86.1) | 351/401 (87.5) | 0.85 | 0.50 | 1.46 | 0.567 |
| | Rarely | | 15/346 (4.3) | 13/401 (3.2) | | | | |
| | Every month | | 21/346 (6.1) | 24/401 (6.0) | | | | |
| | At least once a week | | 12/346 (3.5) | 13/401 (3.2) | | | | |
| Frequency of drinking alcopops‡ | Never | 0.35 | 303/346 (87.6) | 345/402 (85.8) | 1.24 | 0.78 | 1.97 | 0.361 |
| | Rarely | | 19/346 (5.5) | 17/402 (4.2) | | | | |
| | At least once a month | | 24/346 (6.9) | 40/402 (10.0) | | | | |
| Frequency of drinking cider‡ | Never | 0.00 | 305/346 (88.2) | 346/402 (86.1) | 1.26 | 0.80 | 1.97 | 0.313 |
| | Rarely | | 12/346 (3.5) | 13/402 (3.2) | | | | |
| | At least once a month | | 29/346 (8.4) | 43/402 (10.7) | | | | |

*Calculated using the standard $\pi^2/3$ estimator for binary outcomes.
†Single-level Cox regression; percentage of young people who initiated alcohol after baseline; adjusted HR.
‡Multilevel ordinal regression—not consuming 5+ drinks in a row and never drunk anything alcoholic were the reference categories.
ICC, intracluster correlation coefficient; SFP, Strengthening Families Programme.

be delivered as intended, the cultivation of positive group dynamics and the provision of support to participants with additional needs.

### Adverse outcomes
There were no SAEs which were both related (resulting from administration of any of the research procedures) and unexpected.

### DISCUSSION
In this paper we have reported findings from a randomised controlled trial (RCT) of the SFP10-14 following its adaptation for a UK context. We found no evidence of between-group differences 2 years past baseline for the primary outcomes (young people's alcohol consumption and drunkenness in the last 30 days) or for any of the other substance use outcomes. Similarly, there were no between-group differences in relation to young people's well-being, stress and emotional and behavioural problems. Findings were unaltered for preplanned subgroup analyses. Our study findings contrast with evaluations of SFP10-14 conducted in the USA in which long-term effects on substance use outcomes have been detected.[8–10] However, the absence of intervention effects in the UK

**Table 5** Post hoc subgroup analyses for the secondary substance use outcomes and SDQ

| Subgroup | Main effect adjusted OR* | 95% CI Lower limit | 95% CI Upper limit | Subgroup × SFP interaction adjusted OR | 95% CI Lower limit | 95% CI Upper limit |
|---|---|---|---|---|---|---|
| Currently smoke at least one cigarette a week | 1.95 | 0.83 | 4.55 | 0.35 | 0.11 | 1.10 |
| Ever taken cannabis | 2.14 | 0.70 | 6.58 | 0.33 | 0.07 | 1.50 |
| Taken cannabis at least once in the last 12 months | 2.11 | 0.79 | 5.68 | 0.25 | 0.06 | 0.99 |
| Taken cannabis at least once in the last 30 days | 1.22 | 0.39 | 3.77 | 0.30 | 0.06 | 1.54 |

| Subgroup | Adjusted mean difference | 95% CI Lower limit | 95% CI Upper limit | Subgroup × SFP interaction adjusted mean difference | 95% CI Lower limit | 95% CI Upper limit |
|---|---|---|---|---|---|---|
| Strengths and Difficulties total difficulties score | 1.77 | 0.61 | 2.93 | −2.27 | −3.83 | −0.71 |

*The main effect can be interpreted as the effect of belonging to a family with challenges in a group setting (relative to a family without challenges in a group setting) for those allocated to the control arm.

SDQ, Strengths and Difficulties Questionnaire; SFP, Strengthening Families Programme.

setting replicates those of more recent European trials in which adapted versions of SFP10-14 have been evaluated. In Sweden, no effects on substance use or 'breaking behaviours' were detected[27] though significant adaptations made to the intervention may have undermined its hypothesised mechanisms of action.[28] Likewise, findings from the trial conducted in Poland showed no effects on the primary outcomes (substance use) or aspects of family functioning.[29] An RCT in Germany identified only limited evidence of effectiveness.[30] For young people, there were no differences in rates of substance use, except for tobacco initiation (but only among those followed up at 18 months).

Process evaluation findings indicated that SFP10-14UK was delivered with good fidelity (although facilitators' reports were biased towards overstating fidelity), suggesting that this trial was a fair assessment of SFP10-14 when delivered as intended. The PU approach adopted successfully recruited families with differing levels of need (but without the potential stigmatisation generated by targeted interventions), and most groups achieved positive group dynamics in line with hypothesised behaviour change mechanisms.

Several strengths of this study should be noted. There were good participant retention rates and high rates of data completion across both trial arms at main follow-up. At 24-month follow-up there were good levels of agreement between self-report of smoking behaviour and cotinine levels in collected samples. This study was designed as a pragmatic trial in which recruitment, intervention delivery systems and implementation were intended to replicate how the intervention was likely to be delivered outside a trial setting, thus maximising the applicability of our findings. We did not identify evidence of compensatory provision of services in the control arm, increasing our confidence that the study findings would have detected any intervention effects which were present.

In terms of limitations, by virtue of necessity, all outcomes (including the primary outcomes) were measured using self-report. However, self-report of smoking behaviours was validated using measures of salivary cotinine. In line with similar previous evaluations of complex interventions, the trial used multiple statistical tests given the number of intervention outcomes.[31] All outcomes measured hypothesised impacts of SFP10-14UK and mapped on to the intervention's logic model. Again, in line with similar previous studies,[31] we specified in advance the primary and secondary outcomes, and also predefined subgroup analysis, clearly indicating those subgroup analyses which were post hoc.

SFP10-14UK did not replicate the impacts on substance misuse prevention found in the American evaluations. There are a number of possible explanations for the null findings. The American trials tended to have longer term follow-ups and impacts were sometimes stronger in the longer term (eg, over 5–10 years). However, the epidemiological trajectory of young people's drinking, etc, in the UK is earlier than in the US studies so we anticipated any effects to be apparent earlier. The intervention was delivered with good fidelity, and thus implementation failure is unlikely to explain the absence of impacts on hypothesised outcomes. Likewise, cultural adaptations made to SFP10-14 for delivery in the UK prior to this trial focused on aspects of language and presentation of videos, and the core content and functions of the intervention were retained. The intervention as evaluated in this trial is therefore comparable with the original version evaluated in the USA. Trials of SFP10-14 in the USA have been criticised for their approach to data analysis and presentation

of outcomes, with some suggestion that these might have led to unfounded claims for intervention effectiveness.[11 12 32] In this trial, we addressed these criticisms by prespecifying trial outcomes (via a published protocol) and how these would be analysed (through a statistical analysis plan which was approved by an independent steering committee).

It is possible that 'usual care' (to which SFP10-14UK was added for the intervention group) may vary in important ways between the American context and the European settings into which the intervention has been introduced. We were not able to compare data on routine provision of health and welfare services across the current and previous trials. In the present study, SFP10-14UK was integrated into existing parenting, family support and substance misuse prevention provision. Such provision, and their take-up by families, may have varied across settings in which trials of SFP10-14 have taken place. The relatively well-developed suite of services which formed 'usual care' in the current study may have made it more difficult for the intervention to show significant effects.

Another possibility is that intervention content may have interacted in different ways with the context of families receiving it. Post hoc subgroup analysis of substance use-related secondary outcomes appeared to show that for families with challenges in a group setting, differences between intervention and control group were in favour of the intervention group. For families without challenges in a group setting, this pattern was reversed. It is possible that the intervention activities better met the needs of those families identified as likely to experience or present challenges in a group setting, or that key learning had greater fit with their circumstances and was thus easier to adopt and integrate. Alternatively, it may be that group dynamics and the interaction between participants played a role in shaping this patterning, with benefits for those with challenges to the detriment of those without. For example, Wiggins *et al* have previously noted that 'some interventions targeting people at risk can expose participants to the influence of new peers who are more supportive of or more engaged in behaviours associated with risk, thereby spreading risk'.[33] However, no such differences according to families with/without challenges in a group setting were identified for our primary outcomes. While some previous trials of SFP10-14 have identified potential patterning of intervention effects by subgroups according to baseline assessment of risk for later substance use,[30 34 35] our findings here are not directly comparable, since we were concerned with experience/presentation of challenges within a group setting, and not general levels of support needs or risk of later substance use. Previous research on interventions which are targeted based on assessment of young people's risk behaviours has identified the potential for stigmatisation of participants and generation of iatrogenic effects through reinforcement and normalisation of harmful norms.[15 16] The PU approach adopted by implementation of SFP10-14UK in this study appeared to

succeed in reaching families with higher levels of need while avoiding the stigmatisation of participants encountered by targeted interventions. However, it is unclear whether the potential intervention effects for families with challenges and the potential harms for families without challenges may be driven by intervention 'fit', or at least derives partly from group dynamics and the ways in which participants interact.

This trial highlights the importance of assessing the effectiveness of interventions when they are adapted for and then implemented in new settings. Such assessment involves a priori identifying whether an adapted intervention remains faithful to its original logic model, and understanding how it interacts with the new context into which is introduced. This context includes both the existing provision of services, and the needs of the participants who receive it.

**Author affiliations**
[1]Centre for Trials Research, Cardiff University, Cardiff, UK
[2]DECIPHer Centre, School of Social Sciences, Cardiff University, Cardiff, UK
[3]Nuffield Department of Population Health, University of Oxford, Oxford, UK
[4]Department of Psychology, Health and Professional Development, Faculty of Health and Life Sciences, Oxford Brookes University, Oxford, UK
[5]Children's Social Care Research and Development Centre (CASCADE), School of Social Sciences, Cardiff University, Cardiff, UK
[6]College of Human and Health Sciences, Swansea University, Swansea, UK
[7]Centre for Medical Education, School of Medicine, Cardiff University, Cardiff, UK
[8]Swansea Trials Unit, Swansea University Medical School, Swansea University, Swansea, UK
[9]MRC/CSO Social and Public Health Sciences Unit, University of Glasgow, Glasgow, UK

**Acknowledgements** We would like to thank the parents/carers and young people who participated in the trial. Grateful thanks are also due to the delivery teams who implemented the SFP10-14UK as part of the trial, the independent trial steering committee and data monitoring committee, and the Participant Resource Centre at Cardiff University.

**Contributors** JSe wrote the first draft of the paper with input from other authors and led the process evaluation. DG led the design and conduct of statistical analysis and assisted with writing the first draft of the paper and revisions. ML contributed to statistical analysis and the writing of the first draft of the paper and subsequent drafts. JH was a trial manager and collected and analysed the data on recruitment and retention. SM provided input on the design, conduct and analysis of data for the process evaluation. DF advised on the design of the study, the selection of outcome measures and the interpretation of the results. KH contributed to the overall design of the study, the design of the trial's statistical analysis plan and the analysis of trial data. JSc provided input on data collection and analysis within the process evaluation. CP designed and led the health economic evaluation. ZR contributed to the design and conduct of statistical analysis. HR conducted data collection and analysis for the process evaluation. CH was a trial manager and contributed to data collection, and collation of data on recruitment and retention. LM was chief investigator and led the overall design and conduct of the study. All authors contributed to the writing of the paper. JSe is responsible for the overall content as guarantor, accepts full responsibility for the work and the conduct of the study, had access to the data and controlled the decision to publish.

**Funding** Funding of £2.1 million from the National Prevention Research Initiative, managed by the Medical Research Council (award G0802128), included approximately £650 000 implementation costs. The NPRI funding partners are Alzheimer's Research Trust; Alzheimer's Society; Biotechnology and Biological Sciences Research Council; British Heart Foundation; Cancer Research UK; Chief Scientist Office, Scottish Government Health Directorate; Department of Health; Diabetes UK; Economic and Social Research Council; Engineering and Physical Sciences Research Council; Health and Social Care Research and Development Office for Northern Ireland; Medical Research Council; the Stroke Association; Welsh Government; and World Cancer Research Fund. A representative from the

study funders was a member of the trial's independent trial steering committee. The Welsh Government provided approximately £675 000 of partnership funding to cover the cost of implementation in three trial areas, and the associated training and support provided by the Cardiff Strengthening Families Programme team. Further support from the Welsh Government provided £208 000 to cover programme delivery in six trial sites from August 2011 to July 2012. The Cardiff Strengthening Families Programme team also provided financial support for programme delivery and trial recruitment in schools. At the time of the study, DECIPHer was a UKCRC Public Health Research Centre of Excellence. Funding from the British Heart Foundation, Cancer Research UK, Economic and Social Research Council (RES-590-28-0005), Medical Research Council, the Welsh Government and the Wellcome Trust (WT087640MA), under the auspices of the UK Clinical Research Collaboration, is gratefully acknowledged. DECIPHer funding has supported JSe and JH's input into the trial. The centre is now funded by the Welsh Government through Health and Care Research Wales. LM is supported by the Medical Research Council (MC_UU_00022/1) and the Chief Scientist Office (SPHSU16). The Centre for Trials Research is funded by Health and Care Research Wales and Cancer Research UK.

**Competing interests** All authors declare financial support from the National Prevention Research Initiative (managed by the Medical Research Council), Welsh Government and Cardiff Strengthening Families Team. DF's institution has previously received financial support for the development of the SFP10-14UK programme materials from the alcohol industry.

**Patient consent for publication** Not required.

**Ethics approval** This study involves human participants and ethical approval for the trial was given by the Research Ethics Committee for Wales (reference 09/MRE09/53). Participants gave informed consent to participate in the study before taking part.

**Provenance and peer review** Not commissioned; externally peer reviewed.

**Data availability statement** Data are available upon reasonable request. Researchers can request access to deidentified participant data by contacting the corresponding author. Requests for access to data will be subject to approval of a proposal, completion of a signed data access agreement and compliance with institutional and legal requirements. The study protocol is freely available, and the statistical analysis plan is available upon request.

**ORCID iDs**
Jeremy Segrott http://orcid.org/0000-0001-6215-0870
David Gillespie http://orcid.org/0000-0002-6934-2928
Mandy Lau http://orcid.org/0000-0001-5894-570X
Jo Holliday http://orcid.org/0000-0003-4568-7320
Simon Murphy http://orcid.org/0000-0003-3589-3681
David Foxcroft http://orcid.org/0000-0001-9752-7527
Kerenza Hood http://orcid.org/0000-0002-5268-8631
Jonathan Scourfield http://orcid.org/0000-0001-6218-8158
Ceri Phillips http://orcid.org/0000-0003-1076-9289
Laurence Moore http://orcid.org/0000-0003-2182-823X

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
