## [Reviewer comments · BMJ Open]

ARTICLE DETAILS

TITLE (PROVISIONAL)	The effectiveness of the Strengthening Families Programme in the UK at preventing substance misuse in 10-14 year olds: a pragmatic randomised controlled trial
AUTHORS	Segrott, Jeremy; Gillespie, David; Lau, Mandy; Holliday, Jo; Murphy, Simon; Foxcroft, David; Hood, Kerenza; Scourfield, Jonathan; Phillips, Ceri; Roberts, Zoe; Rothwell, Heather; Hurlow, Claire; Moore, L

VERSION 1 – REVIEW

REVIEWER	Moshammer, Hans Medical University of Vienna, Inst Environmental Health, ZPH
REVIEW RETURNED	15-Feb-2021

GENERAL COMMENTS	I very much liked that paper. I am aware of the fact that randomised trials are rare in the field of public health. Therefore, I welcome the study and the paper very much. The study was well conducted and the results are presented clearly and accurately in the paper. I was deeply impressed by it. I nearly wanted to vote "accept" in the first round without further ado. But I still have a small "problem" nagging at my back. Maybe it is my fault or ignorance. I am not sure. I simply do not understand the calculations regarding cost-effectiveness (or what is termed "Headline cost-consequences analysis results" (what ever that might mean). I always thought that for a cost-effectiveness analysis you first need an effect. Then you could quantify the effect per unit of money. But what did you analyse when the intervention seemingly had no effect? In the introduction you promise a cost-effectiveness analysis. I believe what you deliver under results is something different. But to me it is not clear what that really is. Maybe I am not the only one ignorant in the field of costs and economics. Maybe you could try to enlighten these readers?
---

REVIEWER	Ayers, Stephanie Arizona State University Southwest Interdisciplinary Research Center
REVIEW RETURNED	24-Mar-2021

GENERAL COMMENTS	This manuscript provides an interested test of an adapted program to the UK. While the majority of findings are null, this is still an important manuscript for the field. However, there are some concerns with the manuscript. Introduction – The authors' state that "findings of the trials conducted in the USA have been criticised in relation to sample
---

	'selection bias', low rates of intervention attendance, and data analysis and reporting practices." More details of these studies would help strengthen the authors' argument throughout the manuscript as there are many null findings. It is unclear if the authors' have overcome these limitations. Aim – While the authors' clearly describe the overall trial, many of the tertiary outcomes are only provided in the supplemental material and are not discussed anywhere in the manuscript. Because there are so many outcomes and analyses, it was challenging to discern what was an important finding from the Aims of the study. Statistical Analysis – The major concern for this manuscript is the decision on how the authors' arrived at a analytic sample of ~ 746. Given that 931 children provided baseline data, it is unclear why there are only 746 in the analyses, particularly since the authors used multiple imputation. Perhaps this is somewhere in the overwhelming supplemental material, but this should be presented in the main text and justified. Statistical Analysis – It is also necessary to know what the exact attrition rate is and if there were differences between high-risk and low-risk families? Given that there is a subgroup analysis, it is hard to determine if only those high-risk families who contributed data at 2 years were included and perhaps lower functioning high-risk families were not included in the analyses and there are unintended selection biases in the analyses.
--	---

REVIEWER	Newton, Nicola The University of Sydney Faculty of Medicine and Health, Faculty of Medicine
REVIEW RETURNED	29-Mar-2021

GENERAL COMMENTS	The effectiveness of the Strengthening Families Programme in the UK at preventing substance misuse in 10-14 year olds: a pragmatic randomised controlled trial This article describes an important replication trial to evaluate the impact of a UK adapted version of the Strengthening Families Program for young people. There were no differences between the active control and intervention group in most outcomes. Subgroup analysis showed some differences in families with challenges in the control arm compared to families with challenges in the intervention arm. Strength include a large sample size, high retention, a-priori analysis, diverse sample and cost effectiveness analysis. Limitations include the cross-sectional analysis, clarity of writing and limited information on the intervention itself. Abstract:  - Improve clarity of writing and - Include stats method used - Remove mention of outcome in the participants section - Include retention next to retention statistics Introduction:
---

	 - Authors use a proportionate universal approach to surmount challenges of targeted and universal prevention. It would be helpful to include more original citations for the theory behind. More information could be provided on the original theory behind proportionate universalism and ways in which it's implemented as it was unclear how the two components adopted in the current study are different? i.e. 1) recruiting more at-risk families and 2) Having a diverse group. - Consistency around capitalising proportionate universalism or using acronym as mentioned in first instance in the introduction. Methods:  - Writing could be substantially streamlined, particularly in the participants paragraph - Please explain recruitment and 'awareness-raising' in the community more as I wonder whether it's truly universal if recruitment is happening by agencies identifying families that would benefit from the program? E.g. Were there online advertisements / community posters / where were they placed etc. - Could remove table 1 and include in text - Opportunity to provide more information of the intervention itself, including the logic model and key differences made through the adaptation process. What are the 'intervention delivery systems' as referred to in the discussion? - Statistical analysis – unclear where the assumed prevalence numbers are estimated from, citation is for a cost-effectiveness trial of another program. - Unclear why the authors use two level regression models at the 24-month outcome only as opposed to three-level models to estimate intervention by time estimations from baseline and 24months? - Table 3 & 4 – include reference to 24month follow-up in title - Provide mor clarity around how 'likelihood to experience challenges in a group setting' was measured Discussion  - Provide a citation to support the claim that there is better provision of healthcare in the UK compared to US - Opportunity to provide a more in-depth comparison between the US and UK programs to provide insight into the challenges of implementation science and the credibility of the original trial results. E.g. how do the samples differ, how do the programs differ, how did implementation differ etc. Replication is a serious issue in prevention and authors should look to offer future research considerations and recommendations - Authors could specify the strengths and limitations of the current study
--	--

REVIEWER	Sedgwick, Philip St Georges, University of London, Centre for Medical and Healthcare Education
REVIEW RETURNED	20-Apr-2021

GENERAL COMMENTS	Thank you for asking me to review this interesting manuscript. The aim was to investigate the effectiveness of a programme (The Strengthening Families Programme) as applied in the UK at preventing substance misuse in 10-14 year olds. A pragmatic randomised controlled trial study design was adopted.
---

	The manuscript is very well-written and clearly laid out. The study design was suitable and methodology clearly explained. My comments, which are minor, are as follows: Generally, I find the way that results are presented and conclusions made in research articles interesting. This article is no exception but I suspect this is driven by the journal house style more than anything else. For example, the Results section in the Abstract states “There was no evidence of between-group differences 2 years post-randomisation for primary outcomes...”. I would argue there was evidence, but it wasn’t statistically significant. In order to provide clarification, if house style permits I would value inclusion of the phrase “There was no evidence of a statistically significant difference...”. I do wonder if the conclusions might include some suggestion of further studies are warranted, whether it be in different areas of the UK. Based on their sample size calculation, the researchers aimed to recruit and randomise 756 families in total (378 per arm). However, 715 families were eventually recruited into the trial. Is therefore the potential for the trial to be underpowered? Perhaps a comment is warranted (possibly under limitations). The sample size calculation was based on an intracluster correlation coefficient of 0.2. Given the results and the calculated ICCs presented in Table 3, might the ICC used in the sample size calculation be too small? Perhaps a comment is warranted (possibly under limitations). A series of post-hoc subgroup analyses are presented, all of which apparently include statistical hypothesis testing. It is suggested that these should have been exploratory and not involved statistical hypothesis testing to minimise the perils of multiple testing or underpowered comparisons. The researchers indicate in their study proposal published in BMC Public Health that subgroup analyses will be exploratory (and not confirmatory). It was not always clear what the reference category was for each categorical variable in the presented logistic regression models. It may help to add some detail to the tables or further information in a footnote or legend. It was not clear why there are two sets of references (pages # 19 and # 29).
--	--

VERSION 1 – AUTHOR RESPONSE

	Reviewer 1 comments	Response
1	I nearly wanted to vote "accept" in the first round without further ado. But I still have a small "problem" nagging at my back. Maybe it is my fault or ignorance. I am not sure. I simply do not understand the calculations regarding cost-effectiveness (or what is termed "Headline cost-consequences analysis results" (what ever that might mean). I always thought that for a cost-effectiveness	In the methods section of the paper we have inserted more in terms of methods employed and ensured that the terminology used is consistent.

	analysis you first need an effect. Then you could quantify the effect per unit of money. But what did you analyse when the intervention seemingly had no effect? In the introduction you promise a cost-effectiveness analysis. I believe what you deliver under results is something different. But to me it is not clear what that really is. Maybe I am not the only one ignorant in the field of costs and economics. Maybe you could try to enlighten these readers?	
	Reviewer 2 comments	Response
2	Introduction - The authors' state that "findings of the trials conducted in the USA have been criticised in relation to sample 'selection bias', low rates of intervention attendance, and data analysis and reporting practices." More details of these studies would help strengthen the authors' argument throughout the manuscript as there are many null findings. It is unclear if the authors' have overcome these limitations.	In the Introduction section we now include more details on the trials conducted in the USA and provide further details of the methodological criticisms which have been made of them. We also highlight how the present trial addresses some of these methodological limitations.
3	Aim - While the authors clearly describe the overall trial, many of the tertiary outcomes are only provided in the supplemental material and are not discussed anywhere in the manuscript. Because there are so many outcomes and analyses, it was challenging to discern what was an important finding from the Aims of the study.	Our focus in this paper is on the reporting of primary and secondary outcomes, and we acknowledge that pressure of space prevented us from providing detailed information on all trial outcomes. However, the paper does list the tertiary outcomes in the methods section, with the supplementary material providing details of the measures used for each outcome.
4	Statistical Analysis – The major concern for this manuscript is the decision on how the authors' arrived at a analytic sample of ~ 746. Given that 931 children provided baseline data, it is unclear why there are only 746 in the analyses, particularly since the authors used multiple imputation. Perhaps this is somewhere in the overwhelming supplemental material, but this should be presented in the main text and justified.	Our primary analysis set (n=746) included families (and participants within families) in the groups to which they were randomised who provided outcome data. We used multiple imputation and explored other missing data assumptions as part of our sensitivity analysis (n=931). Our sample size calculations planned for loss to follow-up between randomisation and 24-months.
5	Statistical Analysis – It is also necessary to know what the exact attrition rate is and if there were differences between	The attrition rate is similar between the high-risk and

	high-risk and low-risk families? Given that there is a subgroup analysis, it is hard to determine if only those high-risk families who contributed data at 2 years were included and perhaps lower functioning high-risk families were not included in the analyses and there are unintended selection biases in the analyses.	low-risk families at 24-months. 12.2% for the low-risk and 17.1% for the high-risk. We have added this information under the post-hoc subgroup analysis section. As we report, the multiple imputation analyses did not find substantively different results, which would be expected if there was differential non-response bias between the groups
	Reviewer 3 comments	Response
6	Strengths include a large sample size, high retention, a-priori analysis, diverse sample and cost effectiveness analysis. Limitations include the cross-sectional analysis, clarity of writing and limited information on the intervention itself.	In the methods section we have provided additional data about the intervention's theoretical basis under a new sub heading titled 'Intervention'. We used recommended methods for analysing change within randomised controlled trials rather than a cross sectional analysis. We address this in more detail in Row 10 below. We followed a pre-specified data analysis plan that was approved by an independent Study Steering Committee, and that protected against the risk of inflated false positive results.
7	Abstract:  • Improve clarity of writing • Include stats method used • Remove mention of outcome in the participants section • Include retention next to retention statistics 	 • Minor changes made to abstract to improve clarity of writing. • Reference to outcomes in the Participants section removed. • Information on the number of participants is now included in the results section and given the sub heading 'Retention'. • As noted above, we have added additional text on retention. This brings the abstract up to the 300 word limit, and without removing

		essential information from other parts of the abstract we were not therefore able to include additional information statistical methods.
8	Introduction: Authors use a proportionate universal approach to surmount challenges of targeted and universal prevention. It would be helpful to include more original citations for the theory behind. More information could be provided on the original theory behind proportionate universalism and ways in which it's implemented as it was unclear how the two components adopted in the current study are different? i.e. 1) recruiting more at-risk families and 2) Having a diverse group. Consistency around capitalising proportionate universalism or using acronym as mentioned in first instance in the introduction.	 • The term 'Proportionate Universal' is now consistently capitalised throughout. • More detail is provided on the theory behind Proportionate Universalism. We also link this more explicitly to the two components of the approach which were implemented in this study.
9	Methods:  - Writing could be substantially streamlined, particularly in the participants paragraph - Please explain recruitment and 'awareness-raising' in the community more as I wonder whether it's truly universal if recruitment is happening by agencies identifying families that would benefit from the program? E.g. Were there online advertisements / community posters / where were they placed etc. - Could remove table 1 and include in text - Opportunity to provide more information of the intervention itself, including the logic model and key differences made through the adaptation process. What are the 'intervention delivery systems' as referred to in the discussion? - Statistical analysis - unclear where the assumed prevalence numbers are estimated from, citation is for a cost-effectiveness trial of another program. - Table 3 & 4 - include reference to 24month follow-up in title - Provide more clarity around how 'likelihood to experience challenges in a group setting' was measured 	 • Text in the Participants section has been streamlined • We now provide further details about activities to promote awareness raising of the intervention and trial within local communities. • In the methods section we now provide additional information on the process taken to adapt the intervention for a UK context and the key changes made. • In the Methods section (Study Design) we now explicitly refer to delivery systems, so that it is clear to what we are referring back to in the Discussion section. • Prevalence numbers are based on HBSC (Health Behaviour in School Aged Children) survey

		findings (reference 27 in the reference list for the paper).  Title for Tables 3 and 4 now includes reference to 24 month follow-up. Further detail and clarity on how programme staff identified a family as likely to experience challenges in a group setting has been provided.
10	Methods: Unclear why the authors use two level regression models at the 24-month outcome only as opposed to three-level models to estimate intervention by time estimations from baseline and 24months?	Our pre-specified analysis plan included a two-level model with baseline adjusted for as a covariate as the primary analysis so we have to report this as such. While the three-level model as described by the reviewer would allow for the inclusion of a greater number of participants (under a MAR assumption), we chose instead to explore the role of missing observations in a series of sensitivity analyses. Reassuringly, the conclusions drawn from our primary analysis remained in these alternative analyses.
11	Discussion  Provide a citation to support the claim that there is better provision of healthcare in the UK compared to US Opportunity to provide a more in-depth comparison between the US and UK programs to provide insight into the challenges of implementation science and the credibility of the original trial results. E.g. how do the samples differ, how do the programs differ, how did implementation differ etc. Replication is a serious issue in prevention and authors should look to offer future research considerations and recommendations Authors could specify the strengths and limitations of the current study 	 A more nuanced discussion of possible variations in 'usual care' across trial settings is now provided. We extend our comparison of the current study and the previous trials of the intervention within the USA, including how we addressed methodological limitations identified in the USA-based trials, and the comparability of the adapted version of the intervention with the original version.

		 Strengths and limitations of the study are addressed in the discussion.
	Reviewer 4 comments	Response
12	Generally, I find the way that results are presented and conclusions made in research articles interesting. This article is no exception but I suspect this is driven by the journal house style more than anything else. For example, the Results section in the Abstract states “There was no evidence of between-group differences 2 years post-randomisation for primary outcomes...”. I would argue there was evidence, but it wasn’t statistically significant. In order to provide clarification, if house style permits I would value inclusion of the phrase “There was no evidence of a statistically significant difference....”. I do wonder if the conclusions might include some suggestion of further studies are warranted, whether it be in different areas of the UK.	We have amended text throughout from “no evidence of a difference” to “no evidence of a statistically significant difference”.
13	Based on their sample size calculation, the researchers aimed to recruit and randomise 756 families in total (378 per arm). However, 715 families were eventually recruited into the trial. Is therefore the potential for the trial to be underpowered? Perhaps a comment is warranted (possibly under limitations).	While the number of families recruited was lower than planned, follow-up was higher than planned and the total number of families and young people included in our analysis was slightly higher than planned. However, while the ICC for one of our primary outcomes was slightly higher than anticipated, and this may have resulted in a slight net loss in power for this analysis, the effect size and confidence interval excludes our target effect size and thus it is unlikely that our trial suffered from serious power issues.
14	The sample size calculation was based on an intraclass correlation coefficient of 0.2. Given the results and the calculated ICCs presented in Table 3, might the ICC used in the sample size calculation be too small? Perhaps a comment is warranted (possibly under limitations).	See above
15	A series of post-hoc subgroup analyses are presented, all of which apparently include statistical hypothesis testing. It is suggested that these should have been exploratory and not	P-values have been removed from this section and references to statistical

	involved statistical hypothesis testing to minimise the perils of multiple testing or underpowered comparisons. The researchers indicate in their study proposal published in BMC Public Health that sub-group analyses will be exploratory (and not confirmatory).	significance have also been removed.
16	It was not always clear what the reference category was for each categorical variable in the presented logistic regression models. It may help to add some detail to the tables or further information in a footnote or legend.	We have added this information under the table as a footnote.
17	It was not clear why there are two sets of references (pages # 19 and # 29).	One set of references is for the main manuscript. The other set relates to the Supplemental material. We are happy to modify this layout if necessary.

VERSION 2 – REVIEW

REVIEWER	Ayers, Stephanie Arizona State University Southwest Interdisciplinary Research Center
REVIEW RETURNED	05-Oct-2021

GENERAL COMMENTS	Thank you for your responsive comments on this manuscript. However, I am still a bit confused about the sample size. Based on the Consort flow chart (Figure 1), 715 families provided data at baseline, but in the response to the reviewers, it appears the analysis set was based on n=746 families. My assumption is that there are multiple children from the same family that participated in data collection and were included in the analytic sample. If this is the case, the analyses need to adjust for clustering at the family-level. In addition, I did not see the “Multiple imputation was used in a sensitivity analysis” in the Supplementary material.
---

REVIEWER	Newton, Nicola The University of Sydney Faculty of Medicine and Health, Faculty of Medicine
REVIEW RETURNED	08-Oct-2021

GENERAL COMMENTS	My colleague Jennifer Debenham, helped to review this article.
--

VERSION 2 – AUTHOR RESPONSE

We are pleased to submit a revised version of this paper. We have addressed the queries raised by Reviewer 2 as follows:

Query: Thank you for your responsive comments on this manuscript. However, I am still a bit confused about the sample size. Based on the Consort flow chart (Figure 1), 715 families provided data at baseline, but in the response to the reviewers, it appears the analysis set was based on n=746 families.

Response: At baseline, 715 families provided data. A total of 931 young people from these families provided baseline data. At main follow up, 614 families provided data. Data was available for 756 young people. Analysis of the two primary outcomes was 746/732 of the young people.

Query: My assumption is that there are multiple children from the same family that participated in data collection and were included in the analytic sample. If this is the case, the analyses need to adjust for clustering at the family-level.

Response: On Page 8 of the manuscript, we indicate that the analysis accounted for multiple individuals within a family.

Query: In addition, I did not see the “Multiple imputation was used in a sensitivity analysis” in the Supplementary material.

Response: we have now included the information on multiple imputation in the supplementary material.

VERSION 3 – REVIEW

REVIEWER	Ayers, Stephanie Arizona State University Southwest Interdisciplinary Research Center
REVIEW RETURNED	09-Dec-2021
GENERAL COMMENTS	The authors have sufficiently addressed all comments from the reviewer.